# Visual Scene-Aware Hybrid and Multi-Modal Feature Aggregation for Facial Expression Recognition [note 1]

**DOI:** 10.3390/s20185184

**Published:** 2020-09-11

**Authors:** Min Kyu Lee, Dae Ha Kim, Byung Cheol Song

**Affiliations:** Department of Electronic Engineering, Inha University, 100 Inha-ro, Michuhol-gu, Incheon 22212, Korea; lemin0922@gmail.com (M.K.L.); 22171650@inha.edu (D.H.K.)

**Keywords:** facial expression recognition, multi-modal fusion, convolutional neural networks

## Abstract

Facial expression recognition (FER) technology has made considerable progress with the rapid development of deep learning. However, conventional FER techniques are mainly designed and trained for videos that are artificially acquired in a limited environment, so they may not operate robustly on videos acquired in a wild environment suffering from varying illuminations and head poses. In order to solve this problem and improve the ultimate performance of FER, this paper proposes a new architecture that extends a state-of-the-art FER scheme and a multi-modal neural network that can effectively fuse image and landmark information. To this end, we propose three methods. To maximize the performance of the recurrent neural network (RNN) in the previous scheme, we first propose a frame substitution module that replaces the latent features of less important frames with those of important frames based on inter-frame correlation. Second, we propose a method for extracting facial landmark features based on the correlation between frames. Third, we propose a new multi-modal fusion method that effectively fuses video and facial landmark information at the feature level. By applying attention based on the characteristics of each modality to the features of the modality, novel fusion is achieved. Experimental results show that the proposed method provides remarkable performance, with 51.4% accuracy for the wild AFEW dataset, 98.5% accuracy for the CK+ dataset and 81.9% accuracy for the MMI dataset, outperforming the state-of-the-art networks.

## 1. Introduction

Facial expression recognition (FER) is a research topic that is constantly receiving attention in the fields of computer vision and human–computer interaction (HCI). Seven basic emotion categories were first proposed by Ekman and Friesen [1], and then many discrete domain FER studies based on this criterion have been performed. Recently, various methods to recognize the sophisticated emotional intensity in the continuous domain of arousal and valence have been proposed [2]. FER has various applications, such as in-vehicle driver monitoring, facial modeling [3,4], psychology and digital entertainment [5,6]. With recent advances in machine learning, such as deep learning, considerable progress has been made in FER as well as image encryption and fault diagnosis [7,8]. However, robust operation is still difficult because our real-life environment contains various performance degradation factors such as occlusion, complex motion and low illumination. The purpose of this study is to propose an FER algorithm that works robustly even in such wild situations.

Most of the previous studies on FER have adopted discrete emotion categories such as anger, disgust, fear, happiness, sadness, surprise and neutral. Some FER algorithms have even used a continuous domain consisting of valence and arousal. This paper employs universal discrete domain emotion categories.

Early FER techniques focused on classifying human emotions in still images, but research on analyzing emotions of people in videos is becoming popular more and more. There are several datasets for training or verifying video-based FER techniques. For example, the extended Cohn–Kanade (CK+) dataset was collected in a relatively limited environment and used to evaluate the performance of many algorithms such as [9,10,11]. As shown in Figure 1a, CK+ includes videos in which subjects express an artificial emotion. The frontal faces were acquired in an environment with constant illumination, so CK+ cannot be used for designing FER algorithms that should take into account changes of head pose and illumination, as shown in Figure 1b. Thus, robust FER techniques that work well even in wild videos such as Figure 1b are required.

For effective FER in the wild environment, we proposed a visual scene-aware hybrid neural network (VSHNN) that can effectively combine the global and local features from video sequences [12]. The VSHNN consists of a three-dimensional (3D) convolutional neural network (CNN), 2D CNN and RNN. In analyzing the facial features in a video, 3DCNN and 2DCNN are useful for learning temporal information and spatial information, respectively. The RNN can learn the correlation of the extracted features. Meanwhile, it was reported that multi-tasks are more effective in feature analysis than a single task [13]. Thus, this paper assumes that jointly analyzing facial features from multiple aspects can perform much better than analysis from a single aspect. Based on this assumption, this paper analyzes the temporal and spatial information of an input video or their association by using 3DCNN, 2DCNN and RNN, which can be analysis of facial features from various aspects. The 3D CNN with an auxiliary classifier extracts overall spatiotemporal features from a given video. Next, the 2D CNN (fine-tuned DenseNet) extracts local features such as small details from each frame in the video. The RNN properly fuses the two latent features and then final classification is performed. 

To improve the ultimate performance of FER, this paper extends the architecture of the VSHNN and presents a new multi-modal neural network that can exploit facial landmark features as well as video features. To this end, we propose three methods. First, to maximize the performance of RNN in VSHNN, we propose a frame substitution (FS) module that replaces the latent features of less important frames with those of important frames in terms of inter-frame correlation. Second, we propose effective facial landmarks based on a new relationship between adjacent frames. Third, we propose a new multi-modal fusion method that can effectively integrate video and facial landmark information at the feature level. The multi-modal fusion can further improve the performance by applying the attention based on modality characteristics to each modality. The contribution of this paper is summarized as follows.

We propose an FS module to select meaningful frames and show that the FS module can enhance the spatial appearance of video modality during learning.We propose a method to extract features based on facial landmark information to help FER analysis. Because this method reflects correlation between frames, it can extract sophisticated landmark features.We propose an attention-based multi-modal fusion method that can smartly unite video modality and facial landmark modality.

The rest of the paper is organized as follows. Section 2 introduces existing studies on various features and modalities that are used in the proposed method. Section 3 and Section 4 describe the proposed method in detail. Section 5 presents the performance verification of the proposed scheme, and Section 6 concludes the paper. 

## 2. Related Works

### 2.1. Facial Expression Recognition

An FER system recognizes emotions in static images or video sequences by extracting the appearance and geometry features of a person and sensing their changes. The typical features used for conventional FER schemes are mainly hand-crafted features such as the local binary pattern (LBP) [14], scale invariant feature transform (SIFT) [15], histograms of oriented gradients (HOG) [16] and histograms of local phase quantization. Hand-crafted features have also been extended to 3D space–time domains [17,18,19]. Early FER techniques were usually based on these features.

Recently, as deep learning has rapidly developed, FER methods have been using deep learning instead of hand-crafted features. Jung et al. proposed a deep temporal appearance network using a 3D CNN and a deep temporal geometry network based on the multi-layer perceptron (MLP) [20]. Liu et al. proposed an emotional analysis method using a 3D CNN and a deformable part model (DPM) [21]. Mollahosseini et al. modified the well-known Inception network to fit the FER task [22]. Similarly, Hasani et al. proposed an FER scheme using Inception-ResNet and conditional random fields (CRF) [10]. Recently, a generative adversarial network (GAN) [23] was employed for FER. For example, a neutral expression image was generated using GAN, and the residue between a target expression and a neutral expression was learned for FER [9]. 

On the other hand, FER in a daily-life environment with noticeable changes in illuminance or head pose is not an easy task. We can find many FER studies using the still facial expressions in the wild (SFEW) and AFEW datasets [24], which are tailored to wild environments like real life. SFEW is composed of still images, and AFEW is a dataset of videos. Liu et al. presented FER methods using hand-crafted features and CNN features in manifold space and showed emotion analysis results for the AFEW dataset [25,26]. Fan et al. proposed a video-based FER technique based on CNN-RNN and 3D CNN [27]. They also used a network for audio modality and introduced “late fusion” for converging the video network and the audio network at the score level. 

Vielzenf et al. proposed a video-based temporal network using a C3D-LSTM (long short-term memory) structure. They also proposed a late fusion method using a score tree technique and an early fusion using Moddrop [28] to fuse video and audio signals effectively [29]. In addition, transfer learning, in which networks initially learn relatively large datasets and then re-learn a target dataset, was adopted in [27,29,30]. Transfer learning is useful for further improving the performance because FER datasets such as CK+ and AFEW are not large enough for learning.

### 2.2. Feature Correlation

Huang et al. proposed a face recognition method based on the canonical correlation analysis (CCA) of LBP-based 2D texture images and 3D range images obtained from a single image [31]. In the field of object tracking, there have been several studies analyzing object motion and frame consistency based on inter-frame correlation [32,33]. Inspired by these studies, we develop a method of weighting each frame based on inter-frame correlation and use it as a pre-processing module for RNN.

### 2.3. Facial Landmarks

Landmarks are often used as important clues in FER because geometric information about facial changes can be captured through their motion trajectory. Jung et al. converted the trajectories of landmarks into 1D features and applied them to FER [34]. Based on this, Yang et al. transformed landmark trajectories into 2D features and applied them to a CNN [35]. However, the trajectory of landmarks has a disadvantage in that it is hard to extract such information in a wild environment where head poses change rapidly. 

Liu et al. extracted 3D landmarks from videos and defined pairs of 34 points that are closely related to the changes in facial expression [36]. They later proposed a statistical feature based on the Euclidean distances between landmark pairs [37]. Their technique is invariant to head poses because it is based on 3D landmarks and can capture even minute changes in facial expression through Euclidean distance-based statistics.

Conventional techniques for dealing with wild video data have two problems. Firstly, they do not consider the importance of each frame, so less important or unnecessary frames can be included in learning, which may limit the overall performance. Secondly, facial landmark-based methods tend to have noticeably lower performance than image-based ones, which decreases the effectiveness of multi-modal fusion. This paper proposes a solution for each of the above problems and provides a new multi-modal neural network that can effectively fuse various modalities.

## 3. Visual Scene-Aware Hybrid Neural Network

This section proposes an extended architecture of VSHNN. VSHNN is basically a scene-aware hybrid neural network that combines CNN-RNN and 3D CNN to capture global context information effectively [12]. The 3D CNN processes spatial domain information and temporal domain information at the same time, so it encodes the overall context information of a given video. Therefore, fusing the 3D CNN structure with a typical CNN-RNN structure makes it possible to learn based on global context information. In order to select only those frames that are useful for learning video modality, we newly propose a frame substitution (FS) module that replaces the CNN features of less important frames with those of important frames. The importance of a frame is defined as the correlation between features of frames. As a result, only CNN features of meaningful frames are input into the RNN in VSHNN, so global context information can be considered more effectively during the learning process.

As shown in Figure 2, the VSHNN basically follows the CNN-RNN structure. We used DenseNet-FC, which is a well-known 2D CNN and scene-aware RNN (SARNN) as an RNN module. The local features of each frame extracted by DenseNet-FC are called “appearance features,” and the global features extracted by 3D CNN are called “scene features.” VSHNN first extracts appearance features that contain information about small details using the fine-tuned DenseNet-FC, which has learned sufficient emotion knowledge for each frame. Appearance features consist of only features of meaningful frames thanks to the proposed FS module. Next, the 3D CNN extracts scene features consisting of temporal information and spatial information. Finally, emotion classification is performed by using the SARNN to fuse the two types of information effectively.

### 3.1. Pre-Processing

As shown in Figure 2, each piece of input data is pre-processed in advance. The pre-processing for VSHNN consists of face detection, alignment and frame interpolation. Wild datasets such as AFEW include a number of videos with low illumination and occlusion, so we use the multi-task cascaded CNN (MTCNN) [38] for robust face detection. After performing face detection and alignment with MTCNN, the detected area is properly cropped and set as an input image for the following networks. If the input video length is too short to obtain the temporal features, we increase the length of the video to at least 16 frames using a separable convolution-based frame interpolation [39].

### 3.2. Fine-Tuned DenseNet-FC

Recently, transfer learning has been attracting much attention in the field of deep learning as a solution to insufficient training data. If transfer learning is applied, the model learned in the source domain can improve the performance in the target domain [40]. We use a fine-tuned 2D CNN as a transfer learning tool to utilize learned knowledge of facial expression. 

Transfer learning has previously been realized based on the VGG-face model [27,29,30]. Instead of VGG, we use the state-of-the-art (SOTA) classification model DenseNet [41] because it can extract more distinctive features for data analysis. The original DenseNet [41] averages the feature maps through the global average pooling (GAP) layer. To input feature vectors with rich information into the main network, we replace the GAP layer with two fully connected (FC) layers prior to training. This modified network is called DenseNet-FC.

DenseNet-FC is fine-tuned in two steps. First, the DenstNet is pre-trained with the ImageNet dataset [42]. Second, DenseNet-FC is re-trained with the FER2013 dataset [43] to facilitate the analysis of facial expression changes. As a result, the fine-tuned DenseNet-FC can work for any test video in the AFEW dataset at the inference stage. The dimension of the appearance feature is fixed to 4096.

### 3.3. 3D Convolutional Neural Network (3D CNN)

A 3D CNN plays a role in grasping the overall visual scene of an input video sequence because it receives the entire sequence as input. A 3D CNN can be one of three well-known networks: C3D [44], ResNet3D or ResNeXt3D [45]. The 3D CNN is trained via transfer learning. For instance, C3D is usually pre-trained using the Sports-1M dataset [46], and the Kinetic dataset [47] is used for pre-training of ResNet3D and ResNeXt3D. Note that pre-training is performed with action datasets because more dynamic videos provide richer scene features. Then, the pre-trained 3D CNN is fine-tuned, assuming that the convolution layers that serve as filtering in the time–space aspect have already experienced the motion information sufficiently. Therefore, we freeze the convolution layers during fine-tuning. In other words, only the top FC layers are trained so as to refine the scene features in the FC layers.

Next, we add an auxiliary classifier to the last layer of the 3D CNN [48]. The auxiliary classifier makes the learning more stable because of its regularization effect. As in Figure 3, the structure of the auxiliary classifier consists of FC layers, batch normalization [49], dropout and ReLU. Moreover, the softmax function is used in the loss function.

### 3.4. Frame Substitution Module

Conventional FER studies for video sequences [29] employed window-based data pre-processing to select meaningful frames. However, the window-based pre-processing that is basically passive could not adaptively select frames that help FER during learning process. Therefore, we insert an FS module that reflects the statistical characteristics of the video sequence between the fine-tuned DenseNet-FC and the 3D CNN, which makes it possible choose useful frames during the learning process. 

On the other hand, the FS module may inevitably cause duplicate information in the sequence or change the original temporal pattern. However, a facial expression in a video clip occurs only in a specific section of the video clip. In other words, RNN does not consider all the frames of the video clip, but mainly learns only frames that have expressions activated. Therefore, the FS module has a positive effect in terms of performance because it increases the number of activated frames that RNN considers. Moreover, the length of a training purpose video sequence—that is, the time interval for recognizing one facial expression—is usually less than 1 s. During such a short time, different facial expressions seldom exist together. Therefore, it is unlikely that changes in temporal patterns caused by the FS module will have a negative impact on the perception of a certain emotion. Experimental evidence for this is shown in Section 5. The detailed operation of the FS module is as follows.

Let $X={\left\{x_{n}\right\}}_{n=1}^{N}$ be the appearance features obtained from DenseNet-FC (see Figure 2), where $\;x\in R^{D}$. *D* and *N* are the feature dimension and the number of frames in the input video, respectively. To calculate the inter-frame correlation information, we define $Y={\left\{y_{n}\right\}}_{n=1}^{N}$ based on $XX^{T}$, where $\;y\in R^{N}$ is a column vector. Based on **Y**, we calculate the correlation matrix **R**: (1)$$R=\left(\begin{array}{ccc} 1 & \cdots & r_{1N}\\ \vdots & \ddots & \vdots \\ r_{N1} & \cdots & 1 \end{array}\right)$$

In Equation (1), *r_jk_* is the Pearson correlation coefficient between variables $y_{j}$ and $y_{k}$:(2)$$r_{jk}=\frac{s_{jk}}{s_{j}s_{k}}=\frac{\sum_{n=1}^{N}\left(y_{nj}-\bar{y_{j}}\right)\left(y_{nk}-\bar{y_{k}}\right)}{\sqrt{\sum_{n=1}^{N}{\left(y_{nj}-\bar{y_{j}}\right)}^{2}}\sqrt{\sum_{n=1}^{N}{\left(y_{nk}-\bar{y_{k}}\right)}^{2}}}$$
where $\bar{y_{*}}=\frac{1}{N}\sum_{n=1}^{N}y_{n*}$ The value corresponding to the j-th row and the k-th column of R indicates how similar the j-th frame and the k-th frame are among the N frames. An example of this is shown in Figure 4.

Next, we compute the row-wise means of **R** and obtain the frame indexes corresponding to the top *K* means, which are defined as $U^{+}={\left\{u_{n}^{+}\right\}}_{n=1}^{K}$. Similarly, the frame indexes corresponding to the bottom *K* means are defined as $U^{-}={\left\{u_{n}^{-}\right\}}_{n=1}^{K}$. *K* was set to 3 for an input video composed of 16 frames. Note that the *K* value does not damage the temporal structure of the video sequence but substitutes unnecessary frames. Finally, we replace the feature corresponding to the $u_{n}^{-}$-th frame in **X** with the feature corresponding to the $u_{n}^{+}$-th frame, which provides **X**’, which has the same size as **X**. This method helps the RNN to detect facial changes effectively with only a small amount of computation.

### 3.5. Visual Scene-Aware RNN (SARNN)

We propose a temporal network that can fuse global scene features and local appearance features at the feature level. Inspired by a previous study [29], a scene-aware RNN is proposed as an improvement of RNN for the feature-wise fusion of two signals: scene features and appearance features. The scene features from 3D CNN can be used as context because they have temporal information that does not exist in individual frames. We present three types of connections for the efficient fusion of scene features and appearance features (types A, B and C), as shown in Figure 5. Here, we assume that LSTM [50] is used as a model of the RNN. This scene-aware LSTM is called SALSTM.

#### 3.5.1. Type A

Conventional LSTM recognizes the relation between the current and past information by using sequence input $x_{n}$ and hidden state $h_{n}$. However, the proposed SALSTM has a total of three inputs up to the scene feature. In Figure 5a, $x_{n}$ represents an appearance feature, and v represents the scene feature. The appearance feature is input into each unit of SALSTM in temporal order. At the same time, the scene feature is delivered to all units of SALSTM. Thus, the SALSTM is designed to take into account the visual scene, which can be summarized as follows. With an input sequence X, the LSTM is operated as follows:(3)$$i_{n}=\sigma \left(W_{ix}^{T}x_{n}+W_{ih}^{T}h_{n-1}+W_{iv}^{T}v+b_{i}\right)$$
(4)$$f_{n}=\sigma \left(W_{fx}^{T}x_{n}+W_{fh}^{T}h_{n-1}+W_{fv}^{T}v+b_{f}\right)$$
(5)$$o_{n}=\sigma \left(W_{ox}^{T}x_{n}+W_{oh}^{T}h_{n-1}+W_{ov}^{T}v+b_{o}\right)$$
(6)$$g_{n}=\phi \left(W_{gx}^{T}x_{n}+W_{gh}^{T}h_{n-1}+W_{gv}^{T}v+b_{g}\right)$$
(7)$$c_{n}=f_{n}⊗c_{n-1}+i_{n}⊗g_{n}$$
(8)$$h_{n}=o_{n}⊗\phi \left(c_{n}\right)$$

Equations (3)–(6) represent the four gates of SALSTM. $i_{n}$ is an input gate, $f_{n}$ is a forget gate, $o_{n}$ is an output gate, and $g_{n}$ is a candidate gate. In Equations (7) and (8), $c_{n}$ is the cell state, and $h_{n}$ is the hidden state. σ(·) denotes the sigmoid function, and ϕ(·) denotes the hyperbolic tangent function. W is the weight matrix, b is the bias, and ⊗ is the Hadamard product. As a result, $W_{v}$ in Equations (3)–(6) is added to enable feature-wise fusion with the appearance feature as a scene feature clause. The subsequent learning process is equivalent to that of conventional LSTM.

#### 3.5.2. Type B

In general, RNN-based networks initialize all hidden states to zero because they have no previous information ($h_{0}=0$). However, type B makes use of the scene features extracted from 3D CNN as previous information, as shown in Figure 5b. The advantage of such a connection is that temporal information is derived by using the temporally previous information as a whole visual scene, i.e., the scene feature. This can also be easily implemented without changing the LSTM equations. Thus, scene feature **v** can be summarized as follows:(9)$$v=F_{\theta }\left(V\right)$$
(10)$$h_{0}=v$$
where V is the video input, $F_{\theta }$ is the 3D CNN, and θ represents all the training parameters of the 3D CNN.

#### 3.5.3. Type C

As shown in Figure 5c, the simplest method to fuse multi-modal features is feature-wise concatenation, as follows:(11)$$i_{n}=\sigma \left(W_{ix}^{T}\left[x_{n},v\right]+W_{ih}^{T}h_{n-1}+b_{i}\;\right)$$
(12)$$f_{n}=\sigma \left(W_{fx}^{T}\left[x_{n},v\right]+W_{fh}^{T}h_{n-1}+b_{f}\right)$$
(13)$$o_{n}=\sigma \left(W_{ox}^{T}\left[x_{n},v\right]+W_{oh}^{T}h_{n-1}+b_{o}\right)$$
(14)$$g_{n}=\phi \left(W_{gx}^{T}\left[x_{n},v\right]+W_{gh}^{T}h_{n-1}+b_{g}\right)$$

These equations represent the four gates of the LSTM, and the operation [·] is the feature-wise concatenation operation. The last layer $h_{N}$ of the SALSTM plays a role in switching to a probability distribution of emotions via a softmax function prior to classification. Note that the LSTM part of the SALSTM can be replaced with various RNN units, such as a gated recurrent unit (GRU) [51].

### 3.6. Training

The training of VSHNN consists of two steps. In the first step, DenseNet-FC is trained, and the appearance features are extracted on a frame basis. In the second step, 3D CNN and SARNN are trained. As mentioned in Section 3.3, the only FC layers of the pre-trained 3D CNN are fine-tuned in the second step.

We use the cross-entropy function as the loss function of the main network and the auxiliary classifier. Using the cross-entropy loss function, the final loss function $L_{TOTAL}$ is defined as follows: (15)$$L_{TOTAL}=L+\lambda L_{AUX}$$
where L is the main network loss, and $L_{AUX}$ is the loss of the auxiliary classifier. λ is a hyper-parameter that determines the rate of reflection of the auxiliary classifier. The training proceeds such that Equation (15) is minimized.

## 4. Multi-Modal Neural Network Using Facial Landmarks

We need to utilize another modality, i.e., facial landmarks that can be acquired from video, because the performance of FER using only the image modality (pixel information) is limited. Facial landmarks are a sort of hand-crafted feature. Note that FER using only landmark information is inferior to image-based FER [34,35,37,48]. Consequently, multi-modal networks in which the landmark modality is merged with the image modality have been developed. However, the low performance of conventional landmark-based FER techniques is a cause of decreasing the effectiveness of multi-modal fusion. To solve this problem, we enhance the landmark modality and merge the enhanced feature with VSHNN. 

First, we describe the facial landmark in detail, and then we provide three different multi-modal fusion schemes: intermediate concatenation, weighted summation and low-rank multi-modal attention fusion.

### 4.1. Facial Landmark as an Additional Modality

We extend the landmark Euclidean distance (LMED) [37] and propose a new LMED using the correlation between adjacent frames called LMED-CF. LMED is useful for extracting features from wild data, but it is highly dependent on the learning model. Therefore, LMED-CF that is less dependent on the learning model is presented.

The LMED-CF is generated as shown in Figure 6. First, 68 3D landmarks are extracted by [36]. The reason for using 3D landmarks is to cope with extreme head poses that occur frequently in wild data. Next, the Euclidean distance between the landmarks that are closely related to the facial expression change is calculated and transformed into a 34-dimensional feature **I** for each frame ($l\in R^{34}$). This becomes $L={\left\{l_{n}\right\}}_{n=1}^{N}$ for *N*-frame video sequences. In [48], LMED is a 102-dimensional descriptor created by calculating the max, mean and standard deviation of 34-dimensional features in units of frames.

The LMED-CF is defined by adding the inter-frame correlation to the conventional LMED, as shown in Figure 6. The method of computing the inter-frame correlation is discussed in Section 3.4. In detail, **Y** is obtained based on $LL^{T}$, and then a correlation matrix is calculated using Equation (1). Next, an upper triangular matrix excluding the diagonal elements of the correlation matrix is flattened to generate an N(N−1)/2-dimensional feature based on the inter-frame correlation. Finally, four statistics are concatenated and normalized to [−1,1] to construct the final LMED-CF. If *N* is 16, the dimension of LMED-CF is 222 (= 102 + 120). Since LMED-CF provides frame-to-frame or temporal correlation, it can contribute to learning more information than LMED.

### 4.2. Multi-Modal Fusion

This section introduces two representative multi-modal fusion methods and proposes a new multi-modal fusion method that compensates for the shortcomings of the previous methods.

#### 4.2.1. VSHNN-IC: Intermediate Concatenation

Intermediate concatenation (IC) is simply to concatenate features extracted from multiple modality networks. Applying this technique to VSHNN: as shown in the dotted arrow in Figure 7, the scene feature extracted from the 3D CNN and the landmark feature through LMED-CF are concatenated, and then the fused feature information is input into SARNN. Because the two features are information of different modalities, their scales may also be different. To avoid this problem, prior to concatenation, batch normalization (BN) is applied to the scene feature as in Equation (16).
(16)$$v^{\prime }=\left[g\left(v\right),\;l^{\prime }\right]$$
where g(·) and $l^{\prime }$ denote a BN layer and a landmark feature extracted by LMED-CF. The remaining process is the same as in ordinary VSHNN. LMED-CF reflects the statistical characteristics of geometric changes, which can help improve the performance by capturing minute facial expressions that are missed by scene features.

#### 4.2.2. VSHNN-WS: Weighted Summation

As a type of late fusion, weighted summation is a method of properly merging score vectors from heterogeneous networks. For instance, VSHNN-WS simply performs a weighted average on the outputs of networks of different modalities:z = αp + (1 – α)q,   0 ≤ α ≤ 1(17)
where p is the score vector output from VSHNN, q is the score vector output from the landmark-based network, and **z** is the final score. *α* is the weight to be applied to each output and is determined experimentally so that **z** is maximized. In this paper, α was set to 0.6 for AFEW and 0.5 for CK+ and MMI datasets.

#### 4.2.3. VSHNN-LRMAF: Low-Rank Multi-Modal Attention Fusion

The IC scheme is simple but does not fully account for the characteristics of each modality. The WS method is also inefficient because it determines weights manually. Recently, a bilinear pooling based on low-rank decomposition has been proposed to improve the drawbacks of the typical multi-modal fusion methods [52]. However, the performance improvement in [52] is still not so large compared to the IC and WS methods.

Inspired by [52], we propose a new method to further improve the efficiency of low-rank fusion, where self-attention is applied to enhance the feature of each modality. We call it low-rank multi-modal attention fusion (LRMAF). Like VSHNN-WS, LRMAF receives the score vectors from VSHNN and LMED-CF with MLP (see Figure 7). Figure 8 depicts the operation of LRMAF. First, a value of 1 is attached to the last element of each feature vector. Let the output vectors be $z_{vid}\;\mathrm{and}\;z_{lmk}$, respectively. Using the vectors, we calculate self-attentions from Equation (18). Note that the final output is mapped to a probability value between 0 and 1 through the sigmoid function σ(·).
(18)$$a_{vid}=\;\sigma \left(f\left(z_{vid};\varphi_{vid}\right)\right)\;\mathrm{and}\;a_{lmk}=\;\sigma \left(f\left(z_{lmk};\varphi_{lmk}\right)\right)$$
where f(·) indicates a FC layer, and $\varphi_{vid}$ and $\varphi_{lmk}$ stand for the trainable parameters of video and landmark FC layers, respectively.

After element-wise multiplication of the attention and feature vectors corresponding to each modality, take the cross product of the results of the two modalities. Then, the low-rank fusion vector **z** is obtained as in Equation (19).
(19)$$z=(a_{vid}\circ z_{vid})\;⨂\left(a_{lmk}\circ z_{lmk}\right)=z_{vid}^{\prime }⨂\;z_{lmk}^{\prime }$$

Next, in order to reflect the importance of each modality in **z**, we apply low-rank decomposition to the learnable weight matrix **W**. Then, **W** is decomposed into $w_{vid}$ and $w_{lmk}$ as in Equation (20).
(20)$$W≃\sum_{i=1}^{\gamma }w_{vid}^{\left(i\right)}\;⨂\;w_{lmk}^{\left(i\right)}$$
where γ the rank of **W**.

Finally, the output fusion vector $o_{fusion}$ is obtained by applying Equations (19) and (20).
(21)$$o_{fusion}=\left(\sum_{i=1}^{\gamma }w_{vid}^{\left(i\right)}⨂\;w_{lmk}^{\left(i\right)}\right)\cdot \left(z_{vid}^{\prime }⨂\;z_{lmk}^{\prime }\right)=\left(\sum_{i=1}^{\gamma }w_{vid}^{\left(i\right)}\cdot \;z_{vid}^{\prime }\right)\circ \left(\sum_{i=1}^{\gamma }w_{lmk}^{\left(i\right)}\cdot \;z_{lmk}^{\prime }\right)$$

## 5. Experimental Results

### 5.1. Datasets

#### 5.1.1. Acted Facial Expression in the Wild (AFEW)

The videos in the AFEW dataset [53] are difficult to recognize because they are mostly from movies, sitcoms and reality shows in a wild environment rather than an artificial environment. Therefore, the dataset has many videos with low light, head pose variation and severe occlusion. The version of the dataset that we used in the following experiments is AFEW 7.0, which consists of three categories: 773 training data, 383 validation data and 653 test data. Since the dataset is targeted toward the EmotiW challenge, it does not reveal label information for the test set. Therefore, we evaluated the algorithm with the validation data, which have similar tendencies to the test data.

#### 5.1.2. Extended Cohn–Kanade (CK+) Dataset

CK+ [54] is a famous dataset that is often used for the evaluation of FER algorithms. This dataset consists of 593 video sequences of 123 subjects. Only 327 of the 593 videos are labeled with seven emotions (anger, contempt, disgust, happiness, sadness, surprise and neutral). Each video clip begins with a neutral expression and reaches its final frame, which corresponds to an emotional expression. Since the videos were collected in a limited environment, most of the videos are taken from a frontal view, with little change in head pose, and are unnatural because subjects are artificially expressing their emotions.

#### 5.1.3. MMI Dataset

The MMI dataset [55] contains 236 video sequences from 31 subjects of various ages, genders and races. The dataset consists of six labels: anger, disgust, fear, happiness, sadness and surprise. Like CK+, starting from a neural emotion, each sequence was filmed gradually with the expression of a specific emotion label. Some subjects in the MMI dataset have variations in appearance, such as glasses, beards and hats. On the other hand, since the number of data in the MMI is relatively insufficient, MMI is more challenging than CK+. For a fair experiment, we used 205 sequences out of 236 sequences, referring to [56], and adopted 10-fold cross validation as an evaluation method.

### 5.2. Implementation Details

#### 5.2.1. Fine-Tuned DenseNet-FC

We trained the VSHNN on a Xeon E5-2560 and GTX1080Ti. This paper used Ubuntu 16.04 Python 3.6 and PyTorch 1.4.0 for learning. A 16-frame-long video clip was pre-processed in the manner described in Section 3.1 and simultaneously input into the fine-tuned DenseNet and 3D CNN with an auxiliary classifier. We used the DenseNet-121 version in this paper. As mentioned in Section 3.2, all layers of DenseNet121-FC were fine-tuned with the FER2013 dataset. In the DenseNet121-FC, the hidden units of two FC layers were set to a dimension of 4096. The input image size was fixed at 224 × 224 pixels. Data augmentations such as horizontal flipping, rotation, scale and translation were also introduced to enable robust learning. We used SGD with momentum as an optimizer and set the weight decay to 0.0005.

#### 5.2.2. 3D CNN

We used a pre-trained 3D CNN (e.g., C3D, ResNet and ResNeXt) for sequence-based learning. We fixed the weights of the pre-trained network except for the FC layers so that they were not updated during the training process. The number of hidden units of the FC layers in both the main network and auxiliary classifier was set to 2048. The 2048-dimensional scene feature was extracted and used as input for the SARNN. The input image size was set to 112 × 112 pixels, and only horizontal flipping was used for data augmentation. λ was applied to the loss of the auxiliary classifier and experimentally set to 0.5.

#### 5.2.3. SARNN

SARNN receives scene features and appearance features as input. It is unidirectional and uses a single layer. We applied a dropout of 0.5 to prevent overfitting and obtain a regularization effect. We also used gradient clipping to restrict the gradient values to the range of −10 to 10 to prevent “gradient explosion,” which frequently occurs in RNN-based learning. The number of hidden units of SARNN was set to 2048. Finally, the number of hidden units of the FC layers of the SALSTM was set to 1024, and the final prediction score was obtained through the softmax function.

#### 5.2.4. LMED-CF

The 3D landmark points in a detected face image were estimated by a previous method [36]. The total number of frames *N* was set to 16, as in VSHNN. LMED-CF is taught by MLP based on FC layers in the experiments for its validation and late fusion. The MLP structure is based on a module consisting of FC, BatchNorm and ReLU. The number of modules used depends on the datasets. Three modules were used for the AFEW dataset, and two were used for the CK+ and MMI datasets. Detailed configurations of these four neural networks is shown Table A1.

### 5.3. Experimental Results

#### 5.3.1. Effect of Fine-Tuned DenseNet-FC

We evaluated the classification performance of the fine-tuned DenseNet-FC for the FER2013 dataset. In this experiment, we quantitatively compared DenseNet121-FC with VGG16-FACE [29]. “VGG16-FACE (ours)” in Table 1 indicates the numerical value obtained by our experiment. Table 1 shows that DenseNet121-FC provides around 2.3% better performance than VGG16-FACE [29].

We used t-stochastic neighboring embedding (t-SNE) [57] to compare the emotion classification capabilities of the two networks qualitatively. t-SNE is widely used to visualize data by dimensionality reduction of high-dimensional data to low-dimensional data. The 4096-D feature extracted from the last layer (FC7) of VGG16-FACE and the 4096-D feature extracted from the last layer (FC7) of DenseNet121-FC were reduced to a two-dimensional form for effective visualization.

Figure 9a shows that the inter-class boundaries of VGG16-FACE are ambiguous. In certain areas, the anger, fear and neutral clusters are mixed. The DenseNet121-FC result in Figure 9b shows clearer boundaries. We also see the comparison results in terms of the silhouette score, which is used for cluster analysis. Silhouette score is an index commonly used to analyze the consistency of cluster data. In this paper, the silhouette score was employed to evaluate the results of CNN feature analysis. For example, the silhouette score of the *i*-th sample is defined as follows:(22)$$sil\left(i\right)=\frac{\left(b\left(i\right)-a\left(i\right)\right)}{max\left(a\left(i\right),b\left(i\right)\right)}$$
where $a\left(i\right)=\frac{1}{\left|C_{i}\right|-1}\underset{j\in C_{i},i\neq j}{\sum }d\left(i,j\right)$ and $b\left(i\right)=\underset{k\neq i}{min}\frac{1}{\left|C_{k}\right|}\underset{j\in C_{k}}{\sum }d\left(i,j\right)$. a and b indicate the means of intra-cluster distance and nearest-cluster distance, respectively. Moreover, d(i,j) represents the Euclidean distance of the *i*-th and *j*-th sample and $C_{k}$ stands for the number of data in the *k*-th cluster. The closer the silhouette score is to zero, the more ambiguous the boundaries of the cluster are. DenseNet121-FC has a better silhouette score of 0.17 than VGG16-FACE (0.08). We can conclude that the fine-tuned DenseNet-FC extracts appropriate features for FER.

#### 5.3.2. Evaluation of VSHNN on AFEW Dataset

All the networks were learned with the training set of the AFEW dataset and evaluated with the validation set. We examined various combinations of 3D CNN and CNN-RNN to verify the performance of VSHNN. Three 3D CNN models were used for this experiment: C3D, ResNet 3D and ResNeXt 3D. In addition, GRU and LSTM were adopted as the (SA) RNN model, and the performance of three RNN types as shown in Figure 5 was analyzed. For clarity, none of the VSHNN models included the FS module in this experiment.

The left part of Table 2 shows the performance of each single network, and the right part shows the performance of VSHNN, which is a combination of single mode networks. Table 2 shows that the VSHNN models [12] always outperform single models. For example, the best performing network among the single models is the CNN-GRU, with 46.99% accuracy. On the other hand, the lowest performance among the VSHNN models is 47.52% of ResNet 3D and CNN-SAGRU (C). The highest performance is 49.87%, which is the result of the fusion of C3D and CNN-GRU. Note that the synergy effect via fusion becomes around 3% at maximum. This is evidence that VSHNN, a method of fusing global and local features, is more effective for improving FER performance than any single model.

Table 2 also describes the effect of each network on the overall performance of the proposed algorithm. Among the single 3D CNN models, the performance difference between C3D and ResNet 3D amounts to more than 2%. However, the performance variation of VSHNN according to the 3D CNN models is only 1%. This shows that the CNN-RNN model has a greater effect on the overall performance than the 3D CNN model. In addition, Table 2 compares the results for the three connection types. Type B has the best performance among the three types on average, although type A is better than type B for only for C3D and SA-LSTM. Type C tends to be inferior to the others.

On the other hand, we can examine the performance variability according to the sub-network from Table 2. First, to look at the 3D CNN side, the CNN-SARNN model was fixed as CNN-SAGRU (B). Then, the performance of VSHNN is in the range of 48.83% to 49.87% according to the 3D CNN model. That is, it changes by around 1%. Next, the 3D CNN model was fixed to C3D (type B) to observe in terms of CNN-SARNN structure. According to the SARNN model, the performance of VSHNN varies by around 1%. As a result, performance variability due to sub-network changes is not large. In other words, the performance of the proposed scheme is somewhat robust to the perturbation of each sub-network.

Table 3 compares several methods for the AFEW validation dataset. For an exact and fair comparison, the numerical values of the conventional methods were quoted directly from previous studies [27,29,37,58,59,60,61]. CNN-RNN-based techniques [27,29,60] and 2D CNN-based ones [37,58,59,61] were examined. Note that [37] used five-fold cross-validation jointly with a training set and validation set, so it could not be fairly compared with the others. Among the VSHNN models without the FS module, C3D and SAGRU of type B showed higher performance than the previous methods. For example, its accuracy improved by around 2.87% compared to a SOTA method [61].

We performed an additional experiment to examine the effect of the FS module to help improve the performance of RNN. C3D and SA-GRU (B), which showed the best performance in the previous experiment, was also used in this experiment. When applying the FS module, 50.76% accuracy was achieved, which is around 1% better performance than the case without it. This proves that the FS module contributes to the performance improvement by removing less important frames based on inter-frame correlation. Table 4 shows the confusion matrix of C3D and SAGRU (B) with the FS module. It showed better results for anger, happiness and neutral emotions in general but had low performance for disgust and fear. Many emotions tended to be wrongly classified as neutral.

#### 5.3.3. Evaluation of VSHNN on CK+ Dataset

We used CK+ to show that VSHNN works well even for videos in a limited environment. For fair comparison, the numerical values of other methods were quoted exactly from [11,34,62,63]. We used 10-fold validation for a fair experiment. We used seven labels: anger, contempt, disgust, fear, happiness, sadness and surprise, but not neutral. Table 5 shows that C3D and SAGRU (B) provides the best accuracy of 97.25% among VSHNN models without the FS module. C3D and SAGRU (B) is comparable with the SOTA schemes, as shown in the first row of Table 5. As in the AFEW dataset, the performance of C3D and SAGRU (B) was improved by around 0.3% to 97.55% when the FS module was added. Table 6 shows the confusion matrix of C3D and SAGRU (B). The overall performance is good, but the performance for “sad” was somewhat lower than that of other emotions.

#### 5.3.4. Evaluation of VSHNN on MMI Dataset

Table 7 shows the performance of VSHNN for the MMI dataset. For fair comparison, the figures of prior studies are cited in this paper. As in the CK+ dataset, 10-fold cross validation was adopted, and the network structure of C3D and SAGRU (B) with the best performance was used. Firstly, we can see that the FS module improves the performance of VSHNN by around 1.4%. This is evidence that the FS module works for the MMI dataset too. Above all, VSHNN outperforms the conventional techniques. For example, VSHNN outperforms STM-Explet, a typical manifold-based algorithm, up to around 3.5%, and has around 1.6% higher performance than MSCNN [56], which is one of the SOTA deep neural networks using distance metric-based loss. In addition, Table 8 shows the confusion matrix for the VSHNN with FS module. Happiness and surprise showed high figures of 90.2% and 89.4%, respectively, while fear showed the lowest accuracy of 30%. As can be seen from the confusion matrix, fear and surprise are difficult to distinguish from each other, so other SOTA techniques seldom distinguish fear emotions.

#### 5.3.5. Evaluation of the LMED-CF

The effect of LMED-CF on FER was validated here. Similar to the previous experiment, we used the validation set of the AFEW dataset and the 10-fold cross-validation for the CK+ dataset. Table 9 shows that LMED is valid in the SVM model, but its performance deteriorates in the MLP model, as mentioned in Section 3.6. LMED-CF with inter-frame correlation showed better performance than LMED in both models.

LMED-CF showed almost the same performance as a SOTA scheme, the deep temporal geometry network (DTGN) [34], in the CK+ dataset, but it significantly outperformed DTGN in the AFEW dataset. This result is caused by the fact that the 2D landmark trajectory used by DTGN was not formed effectively for videos with severe head-pose variation. LMED-CF is relatively robust to head pose because it is based on 3D landmarks. In particular, the accuracy of LMED-CF with the MLP model is 42.56%, which is around twice that of LMED and DTGN using the same model.

An additional experiment was performed on the CK+ dataset to analyze the noise-robustness of LMED-CF. By incorporating additive white Gaussian noise (AWGN) into landmark points extracted from CK+, we intentionally degraded the accuracy of the landmark information. At this time, the standard deviation of AWGN was increased by 0.5 increments from 0 to 3, and the experimental environment was consistent with that of Section 5.2.4. Figure 10 shows the results of this experiment. In the absence of noise, the performance is 92.38%, as shown in Table 9. As the noise increases, the performance decreases. However, there was no sudden drop in performance. Especially, it is notable that LEMD-CF maintains an average of 90% or more even in [14,44], despite such noise significantly interfering with the accuracy of the landmark position. This proves that the proposed LMED-CF is robust to noise.

#### 5.3.6. Multi-Modal Fusion

This section analyzes the performance of various multi-modal fusion methods. A total of four different fusion methods were applied to the VSHNN framework: VSHNN-IC (see Section 4.2.1), VSHNN-WS (see Section 4.2.2), VSHNN-LRMF and VSHNN-LRMAF (see Section 4.2.3). Note that LRMAF is the proposed multi-modal fusion method, and low-rank multi-modal fusion (LRMF) is taken from [52]. The same three datasets were used as in the previous experiments. For relative comparison with the SOTA performance of multi-modal fusion, Moddrop [29] was adopted for the AFEW dataset and MSCNN-PHRNN [56] was benchmarked for the remaining datasets. MSCNN-PHRNN is multi-modal fusion using landmark and video modality like LRMAF. However, Moddrop adopted audio modality as well as video modality while using more than two sub-networks per modality.

Table 10 shows the experimental results for the AFEW dataset. WS had the lowest performance of 50.13%. This performance is less than the basic VSHNN, which uses only video modality. IC and LRMF show 0.06% and 0.15% improvement over the basic VSHNN, respectively. We can see that LRMF which considers element-wise inter-modality correlation is marginally better than IC. Note that LRMAF is 0.67% higher than the basic VSHNN and also 0.52% higher than LRMF. The main reason for such a performance improvement is that feature enhancement according to attention per modality has a positive effect on low-rank fusion. As another point of view, take a look at the computational time. When measuring the computational times for training, all the methods in Table 10 showed similar times within the range of 4 to 6 h. This means that the time it takes for the learning parameters to be optimized is similar regardless of the network size. Moreover, to analyze the computational time for testing, we measured the inference time. Here, the inference time indicates the total processing time taken from the data input. The backbone, i.e., VSHNN, took 1.8 ms based on a batch size of 32. IC and WS required computational times similar to the naive backbone. However, low-rank fusion methods such as LRMF and LRMAF consumed considerable computational times because computationally heavy matrix multiplication is required to calculate the low-rank fusion vector as in Equation (19). Thus, one of our future goals is to develop a lightweight fusion network.

Table 11 shows the confusion matrix of VSHNN-LRMAF. The two highest performing emotions were anger and happiness, 85.9% and 74.6%, respectively. Compared with Table 9, anger and sadness became around 20% and 8.2% better, respectively. Table 12 shows the results for CK+. VSHNN-LRMAF shows the best performance among VSHNN methods and provides almost the same performance as MSCNN-PHRNN, a SOTA technique. In fact, 98.5% may be the upper boundary in performance. Finally, Table 13 shows the results for the MMI dataset. LRMAF is around 4% better than the single modality method and 0.75% higher than MSCNN-PHRNN.

### 5.4. Discussion

#### 5.4.1. Intuition of Model Selection

First, referring to [29], the FC layer dimensions of DenseNet-FC and 3D CNN were set to 4096 and 2048, respectively. This setting is somewhat disadvantageous in terms of the computational complexity, but it is advantageous in learning facial expression information through more learning parameters. The same is true of SARNN’s hidden unit. Next, three models, C3D, ResNet 3D and ResNeXT 3D, were selected as 3D CNN models because they showed higher performance in the AFEW dataset (see Table 2). Lastly, the reason for varying the number of FC layers according to the dataset in LMED-CF was to consider the difference in difficulty between datasets. For example, the AFEW dataset is classified as a wild dataset because the lighting change or occlusion is more severe than other datasets. Therefore, FC layers were added so that the proposed method could work well even in wild datasets with high difficulty. Other parameters required for learning were determined through several experiments.

#### 5.4.2. Applicability of the Proposed Method

The FS module of the proposed method finds a frame that is useful for emotion analysis through the similarity between frames, and then the similarity between facial landmarks in the selected frames is analyzed. In the future, several skills in our approach can be applied to analyze hand gestures and body actions. For example, when recognizing hand gestures, the correlation (or covariance) between the hand features and the non-hand features enables the model to be robust against outliers or occlusions around hand(s).

## 6. Conclusions

This paper proposed an extended architecture of VSHNN [12] and a new multi-modal neural network for improving the FER performance ultimately. Specifically, we proposed an FS module that replaces less important frames in a video sequence with important frames and a method to extract facial landmark-based features through the correlation between frames to achieve the SOTA performance of FER. In addition, a new attention-based low-rank bilinear pooling was proposed to maximize the synergy between video and landmark information. The experimental results showed that the proposed multi-modal network can provide significant accuracy of 51.4% for the wild AFEW dataset, 98.5% for the CK+ dataset and 81.9% for the MMI dataset.

## Figures and Tables

**Figure 1 sensors-20-05184-f001:**
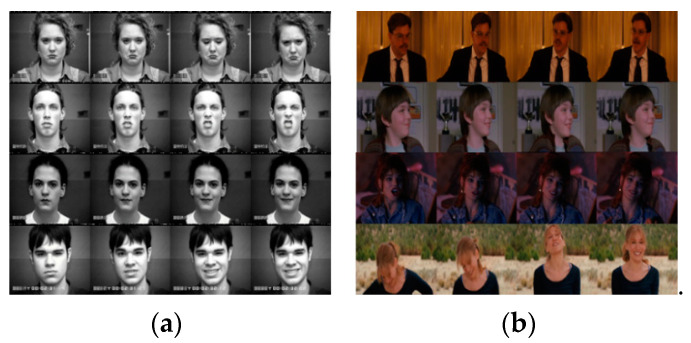
(**a**) CK+ dataset in a limited environment; (**b**) AFEW dataset in a wild environment.

**Figure 2 sensors-20-05184-f002:**
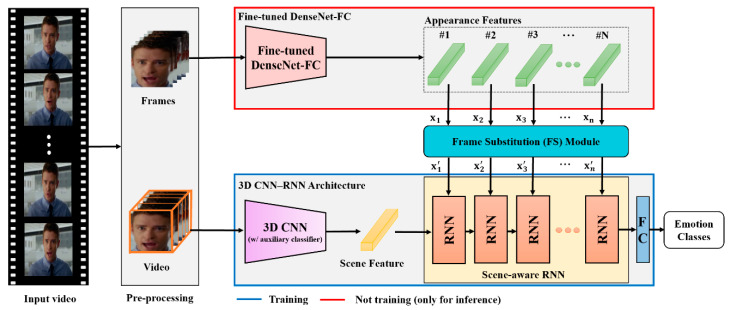
Overview of the extended VSHNN architecture.

**Figure 3 sensors-20-05184-f003:**
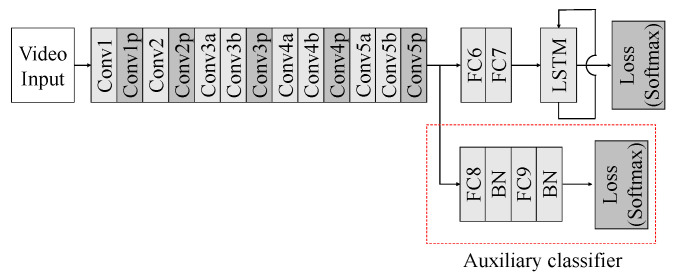
Structure of 3D CNN with auxiliary classifier.

**Figure 4 sensors-20-05184-f004:**
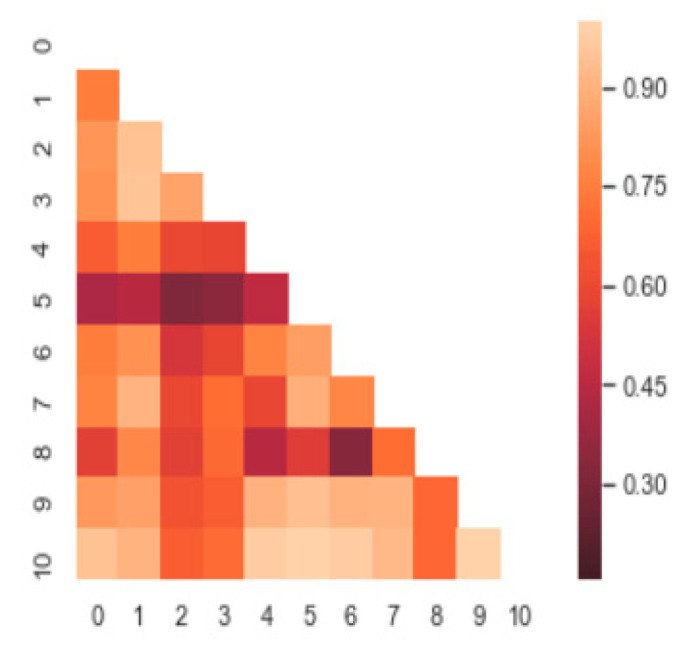
Example of correlation matrix.

**Figure 5 sensors-20-05184-f005:**
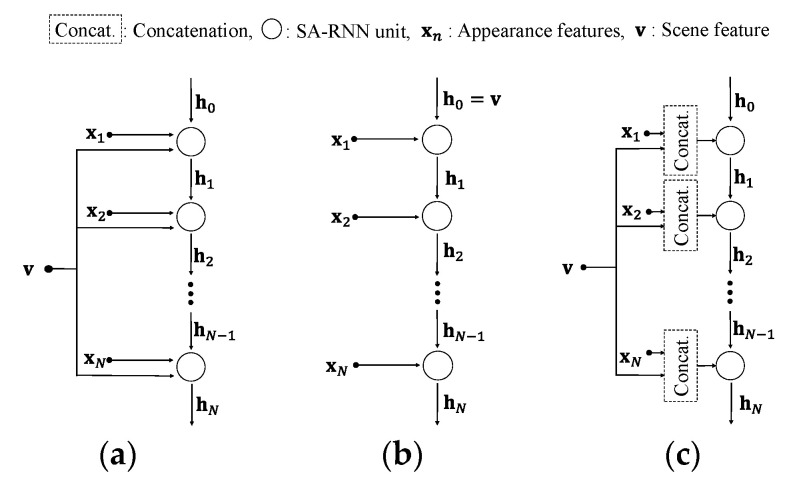
Three types of visual scene-aware LSTM. (**a**) Type A, (**b**) Type B, (**c**) Type C.

**Figure 6 sensors-20-05184-f006:**
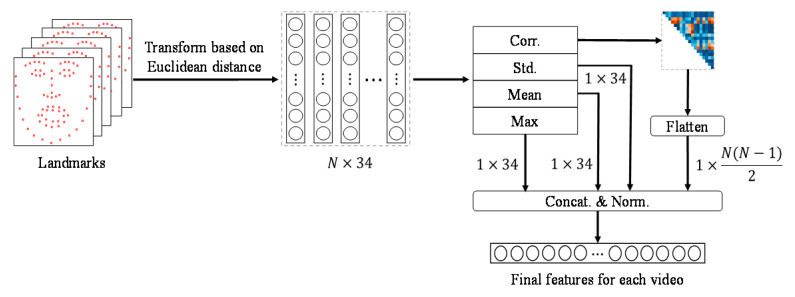
Generation of LMED-CF. Euclidean distance-based transformation is used [37].

**Figure 7 sensors-20-05184-f007:**
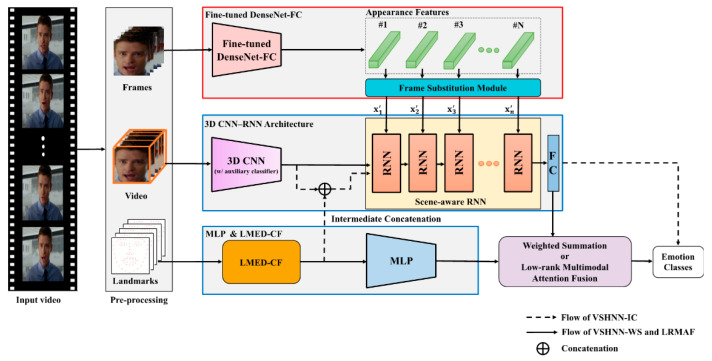
Block diagram of VSHNN-LRMAF.

**Figure 8 sensors-20-05184-f008:**
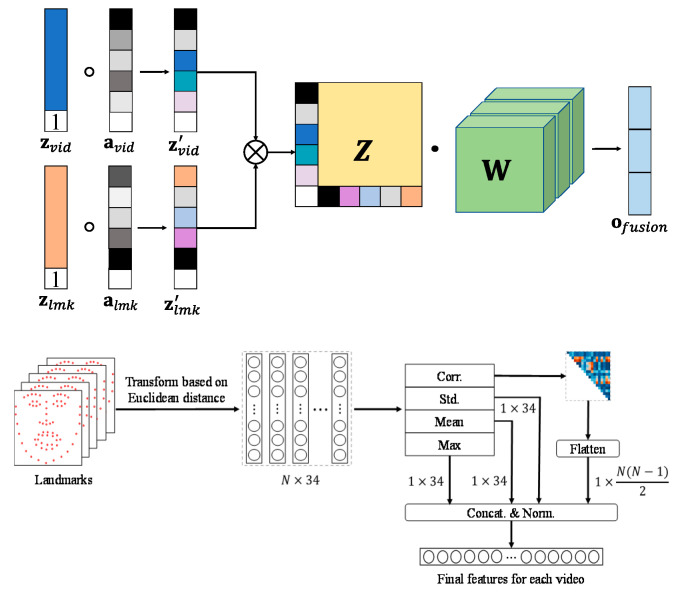
The operation of low-rank multi-modal attention fusion (LRMAF).

**Figure 9 sensors-20-05184-f009:**
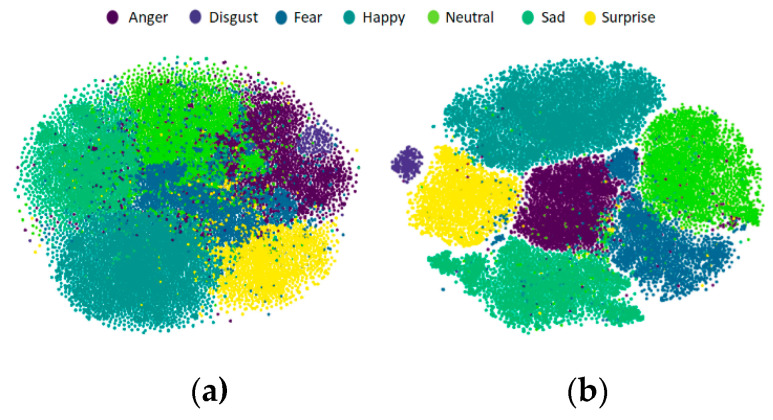
Visualization study of (**a**) VGG-16 FACE and (**b**) DenseNet121-FC.

**Figure 10 sensors-20-05184-f010:**
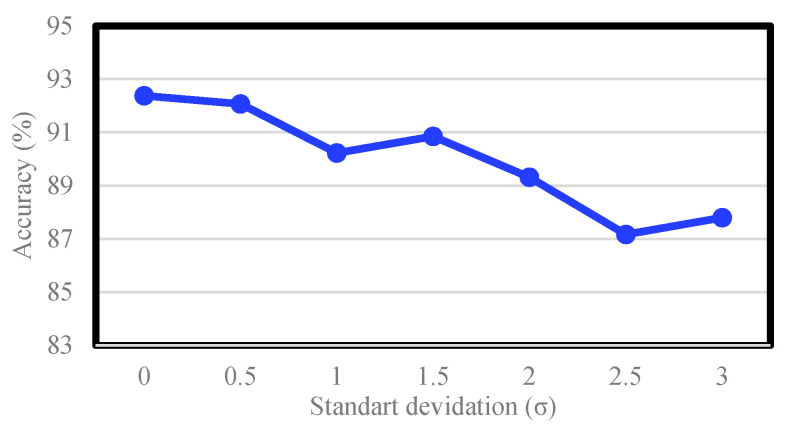
Accuracy of LMED-CF according to the noise standard deviation.

**Table 1 sensors-20-05184-t001:** Performance comparison of supervised learning for MAHNOB-HCI.

Method	Average Acc. (%)	Silhouette
VGG16-FACE [29]	71.20	-
VGG16-FACE (ours)	72.27	0.08
DenseNet121-FC	73.46	0.17

**Table 2 sensors-20-05184-t002:** Performance evaluation of various models for AFEW dataset. VSHNN without frame substitution module indicates [12]. The CNN in conventional single models is DenseNet121-FC. Bold indicates the best of each row.

Conventional Single Models	Accuracy (%)	VSHNN without Frame Substitution Module [12]	Accuracy (%)		
			A	B	C
C3D	41.25	C3D and CNN-SALSTM	49.61	48.83	48.04
ResNet 3D	39.16	C3D and CNN-SAGRU	48.56	49.87	48.04
ResNeXt 3D	41.01	ResNet3D and CNN-SALSTM	48.56	48.56	47.78
CNN-LSTM	46.74	ResNet3D and CNN-SAGRU	48.56	48.83	47.52
CNN-GRU	46.99	ResNeXt3D and CNN-SALSTM	48.30	48.83	47.78
		ResNeXt3D and CNN-SAGRU	48.56	49.09	48.04

**Table 3 sensors-20-05184-t003:** Evaluation results for AFEW dataset. The letters in parentheses indicate the types of CNN-SARNN, and * means that the experimental method is five-fold cross-validation. VSHNN without FS module indicates [12].

	Method	Accuracy (%)
Conventional methods	SSE-HoloNet [59]	46.48
	VGG-LSTM [27]	45.42
	VGG-LSTM [29]	48.60
	C3D-LSTM [29]	43.20
	DSN [58]	48.04
	ResNet-50+BLSTM [60]	49.31
	Av. Pool (1) [61]	49.70
	Densenet-161 pool5 *** [37]	51.17
VSHNN without FS [12]	C3D and SALSTM (A)	49.61
	C3D and SAGRU (B)	49.87
VSHNN with FS	C3D and SAGRU (B)	50.76

**Table 4 sensors-20-05184-t004:** The confusion matrix of VSHNN with FS module for AFEW dataset.

	Ang.	Dis.	Fear	Hap.	Neu.	Sad	Surp.
Ang.	65.6	0.0	7.8	1.6	12.5	3.1	9.4
Dis.	12.5	7.5	7.5	7.5	35.0	17.5	12.5
Fear	15.2	4.4	15.2	6.5	21.7	26.1	10.9
Hap.	1.6	0.0	1.6	77.8	7.9	9.5	1.6
Neu.	3.2	3.2	3.2	3.2	77.8	6.4	3.2
Sad	1.6	1.6	1.6	9.8	29.5	39.3	16.4
Surp.	8.7	0.0	10.87	6.5	21.7	4.4	47.8

**Table 5 sensors-20-05184-t005:** Comparison of average accuracy for CK+ dataset. The letters in parentheses indicate the types of SARNN. VSHNN without the FS module corresponds to [12]. * Indicates a method using only image information.

	Methods	Accuracy (%)
Conventional methods	STM-Explet [63]	94.19
	DTAGN [34]	97.25
	PPDN [62] *	97.30
	DeRL [9] *	97.30
	HNwFL [11]	97.35
VSHNN without FS [12]	ResNet3D and SALSTM (A)	96.63
	ResNeXt3D and SAGRU (B)	96.96
	C3D and SAGRU (B)	97.25
VSHNN with FS	C3D and SAGRU (B) with FS	97.55

**Table 6 sensors-20-05184-t006:** The confusion matrix of VSHNN with FS module for CK+ dataset.

	Ang.	Cont.	Disg.	Fear	Hap.	Sad	Surp.
Ang.	97.8	0.0	0.0	0.0	0.0	2.2	0.0
Cont.	0.0	94.4	0.0	0.0	5.6	0.0	0.0
Disg.	1.7	0.0	98.3	0.0	0.0	0.0	0.0
Fear	0.0	0.0	0.0	100.0	0.0	0.0	0.0
Hap.	0.0	0.0	0.0	0.0	100.0	0.0	0.0
Sad	13.8	0.0	0.0	0.0	0.0	86.2	0.0
Surp.	0.0	1.2	0.0	0.0	0.0	0.0	98.8

**Table 7 sensors-20-05184-t007:** Comparison of average accuracy for MMI dataset. The letters in parentheses indicate the type of SARNN. * indicates a method using only image information.

	Methods	Accuracy (%)
Conventional methods	STM-Explet [63]	75.12
	DTAGN [34]	70.24
	DeRL [9] *	73.23
	MSCNN [56]	77.05
VSHNN without FS	C3D and SAGRU (B)	77.21
VSHNN with FS	C3D and SAGRU (B) with FS	78.60

**Table 8 sensors-20-05184-t008:** The confusion matrix of VSHNN with FS module for MMI dataset.

	Ang.	Dis.	Fear	Hap.	Sad	Surp.
Ang.	75.76	0	3.03	0	12.12	9.09
Dis.	9.09	81.82	0	9.09	0	0
Fear	5.0	5.0	70.0	0	15.0	5.0
Hap.	0	4.88	4.88	90.24	0	0
Sad	6.45	6.45	9.68	0	74.19	3.23
Surp.	4.26	0	17.02	2.13	0	76.60

**Table 9 sensors-20-05184-t009:** Comparison with landmark-based networks.

Method	Dataset	Model	Accuracy (%)
LMED [37]	AFEW	SVM	39.95
LMED [37]		MLP	20.89
DTGN [34]		MLP	21.02
LMED-CF		SVM	40.99
LMED-CF		MLP	42.56
DTGN [34]	CK+	MLP	92.35
LMED-CF		MLP	92.38

**Table 10 sensors-20-05184-t010:** Performance evaluation of multi-modal fusion for AFEW dataset. * Note that Moddrop uses an additional audio modality. Note that interference time was measured based on batch size of 32.

Method	Inference Time (ms)	Mode	Accuracy (%)
VSHNN	1.8	Single mode (video)	50.76
VSHNN-IC	4	Multi-mode (Landmark + video)	50.82
VSHNN-LRMF	278		50.91
VSHNN-LRMAF	282		51.43
VSHNN-WS	5		50.13
Moddrop [29] *	-	Multi-mode (Audio + video)	52.20

**Table 11 sensors-20-05184-t011:** The confusion matrix of VSHNN-LRMAF for AFEW dataset.

	Ang.	Dis.	Fear	Hap.	Neu.	Sad	Surp.
Ang.	85.9	4.6	3.1	1.6	1.6	1.6	1.6
Dis.	22.5	10.0	7.5	5.0	27.5	22.5	5.0
Fear	34.8	2.2	15.2	6.5	17.4	19.6	4.4
Hap.	3.2	1.6	1.6	74.6	7.9	9.5	1.6
Neu.	6.4	0.0	1.6	7.9	65.1	12.7	6.4
Sad	8.2	1.6	1.6	6.6	27.9	47.5	6.6
Surp.	17.4	2.2	10.9	6.5	19.6	13.0	30.4

**Table 12 sensors-20-05184-t012:** Performance evaluation of multi-modal fusion for CK+ dataset.

Method	Mode	Accuracy (%)
VSHNN	Single mode (video)	97.55
MSCNN-PHRNN [56]	Multi-mode (Landmark + video)	98.50
VSHNN-IC		97.86
VSHNN-LRMF		98.27
VSHNN-LRMAF		98.48
VSHNN-WS		98.07

**Table 13 sensors-20-05184-t013:** Performance evaluation of multi-modal fusion for MMI dataset.

Method	Mode	Accuracy (%)
VSHNN	Single mode (video)	78.60
MSCNN-PHRNN [56]	Multi-mode (Landmark + video)	81.18
VSHNN-IC		76.10
VSHNN-LRMF		81.60
VSHNN-LRMAF		81.93
VSHNN-WS		79.02

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
