# Peer review of "Visual Scene-Aware Hybrid and Multi-Modal Feature Aggregation for Facial Expression Recognition†"

_sensors, 2020, doi:10.3390/s20185184_

Round 1

Reviewer 1 Report

The authors propose an approach for FER by combining deep learning networks and fusing features. The topic is interesting and the paper contributes to the field of FER.

My comments on the current paper are the following.

1. The references are not arranged logically. The authors should adjust the orders of the references according to the style of the journal.
2. L55-L60. The reason for choosing 3DCNN,2DCNN and RNN is not clearly presented. The novelty or innovation should be highlighted in the Introduction.
3. Convolution/filtering plays an important role in the paper. Besides feature extraction for FER, it can be used in prediction, encryption, fault diagnosis (eg.,10.3390/en12183560, 10.3390/e21030319, 10.1016/j.ijepes.2018.03.001), and so on. The authors may introduce or cite these works or some others to show the power of convolutional operations.
4. Silhouette score is not commonly used in classification, so it would be better to give its mathematical equations in the paper.
5. In addition to the hardware environment, what programming language or DL software platform is used in this paper? It is useful for readers to reproduce the experiments.
6. The paper mainly uses Accuracy to evaluate the methods. We know that Accuracy is usually not enough to evaluate classification methods. The authors should consider using more evaluation metrics to evaluate FER methods in the experiments.
7. Since the proposed approach combines several DL networks and several types of features, how about the time complexity for training and testing? What does "Inference time" mean in Table 10?
8. There are a lot of parameters in Sec. 5.2, it would be better to list them in a table.

some minor issues:

L585. ... interference time...

Author Response

I enclose the replies to the reviewers' comments as one file. 

Reviewer 2 Report

 The article deals with Facial Expression Recognition and proposes a novel architecture that effectively fuses image and landmark information. In practice the article proposes an extended architecture of visual scene-aware hybrid neural network (VSHNN) that consists of three dimensional (3D) convolutional neural network (CNN), 2D CNN, and RNN. Initially a frame substitution module is proposed that replaces the latent features of less important frames with those of important frames. Secondly a method is proposed for extracting facial landmark features based on the correlation between frames. Finally, a new attention-based low-rank bilinear pooling is proposed to maximize the synergy between video and landmark information. Moreover, the authors depicted a set of detailed experimental results that prove the validity of their claims and the superiority of their approach.

The article is interesting and well written. Moreover, the algorithmic schemes employed are novel and well structured.  Therefore, I propose acceptance subject to minor revision. I propose minor revision because I would like to see in the article more discussion concerning the intuition behind the choice of the various algorithmic parameters and the combination of the algorithmic components, plus a discussion concerning using the techniques to other applications too, behind Facial Expression Recognition. Explanation of the intuition is imperative when presenting the details of the neural network employed.  

Author Response

The article deals with Facial Expression Recognition and proposes a novel architecture that effectively fuses image and landmark information. In practice the article proposes an extended architecture of visual scene-aware hybrid neural network (VSHNN) that consists of three dimensional (3D) convolutional neural network (CNN), 2D CNN, and RNN. Initially a frame substitution module is proposed that replaces the latent features of less important frames with those of important frames. Secondly a method is proposed for extracting facial landmark features based on the correlation between frames. Finally, a new attention-based low-rank bilinear pooling is proposed to maximize the synergy between video and landmark information. Moreover, the authors depicted a set of detailed experimental results that prove the validity of their claims and the superiority of their approach.
The article is interesting and well written. Moreover, the algorithmic schemes employed are novel and well structured. Therefore, I propose acceptance subject to minor revision.

1. I propose minor revision because I would like to see in the article more discussion concerning the intuition behind the choice of the various algorithmic parameters and the combination of the algorithmic components, plus a discussion concerning using the techniques to other applications too, behind Facial Expression Recognition. Explanation of the intuition is imperative when presenting the details of the neural network employed.

Response:
According to the reviewer’s comment, we discussed the model selection issue in Section 5.4 of the revised manuscript, and mentioned the possibility of applying the proposed method to other research fields.
New:
5.4.1. Intuition of Model Selection
First, referring to [29], the FC layer dimensions of DenseNet-FC and 3D CNN were set to 4096 and 2048, respectively. This setting is somewhat disadvantageous in terms of the computational complexity, but it is advantageous in learning facial expression information through more learning parameters. The same is true of SARNN’s hidden unit. Next, three models, C3D, ResNet 3D, and ResNeXT 3D, were selected as 3D CNN models because they showed higher performance in the AFEW dataset (see Table 2). Lastly, the reason for varying the number of FC layers according to the dataset in LMED-CF was to consider the difference in difficulty between datasets. For example, the AFEW dataset is classified as a wild dataset because the lighting change or occlusion is more severe than other datasets. So, FC layers were added so that the proposed method can work well even in wild datasets with high difficulty. Other parameters required for learning were determined through several experiments.
5.4.2. Applicability of the Proposed Method
The FS module of the proposed method finds a frame that is useful for emotion analysis through the similarity between frames, and then the similarity between facial landmarks in the selected frames are analyzed. In the future, several skills in our approach can be applied to analyze hand gestures and body actions. For example, when recognizing hand gestures, the correlation (or covariance) between the hand features and the non-hand features enables to be robust against outliers or occlusions around hand(s).

Reviewer 3 Report

Facial expression recognition (FER) is a trendy topic of interest for the readership of the journal. This paper delas with two main goals desbribed as follows:i) a frame substitution module that replaces the latent features, ii)a method for extracting facial landmark features based on the correlation between frames.

I am very positive on research in this front, and I would like to congratulate the authors on the proposal. I find no objective errors, and I personally would like to read more on the applications to the different fields, as described at the introduction.

I only have minor points related to its theoretical basis and applications.

First of all, I missed more information on the selection of emotion. I imagine authors have selected the universal emotions according Ekman's theory. So, I suggest to explain it in a more extended way at the introduction and the methodology.

Moreover, I find the manuscript could be improved enphasizing its applications in the different fields, not only naming them. It could raise the interest from them and increase the manuscript impact. As a suggestion in neuroscience, which is my area of research , I would cite the following papers:

i) for healthy populations, the importance of face discrimination 

Moret-Tatay, C., Baixauli-Fortea, I., & Grau-Sevilla, M. D. (2020). Profiles on the Orientation Discrimination Processing of Human Faces. International Journal of Environmental Research and Public Health17(16), 5772.

ii) and clinical ones

Storey, G. (2019). Deep human face analysis and modelling (Doctoral dissertation, Northumbria University, Newcastle upon Tyne, UK).

Author Response

Facial expression recognition (FER) is a trendy topic of interest for the readership of the journal. This paper deals with two main goals described as follows: i) a frame substitution module that replaces the latent features, ii) a method for extracting facial landmark features based on the correlation between frames.
I am very positive on research in this front, and I would like to congratulate the authors on the proposal. I find no objective errors, and I personally would like to read more on the applications to the different fields, as described at the introduction.
I only have minor points related to its theoretical basis and applications.1. First of all, I missed more information on the selection of emotion. I imagine authors have selected the universal emotions according Ekman's theory. So, I suggest to explain it in a more extended way at the introduction and the methodology.Response:
The recommended information about the discrete emotion category was additionally dealt with in Lines 32-35 of the revised manuscript. On the other hand, some methods for analyzing the emotional intensity in the arousal/valence domain have been recently proposed. So, we described this research trend together as follows.
New:
“Seven basic emotion categories were first proposed by Ekman and Friesen [1], and then many discrete domain FER studies based on this criterion have been performed. Recently, various methods to recognize the sophisticated emotional intensity in the continuous domain of arousal and valence have been proposed [2].”

2. Moreover, I find the manuscript could be improved enphasizing its applications in the different fields, not only naming them. It could raise the interest from them and increase the manuscript impact. As a suggestion in neuroscience, which is my area of research, I would cite the following papers:
i) for healthy populations, the importance of face discrimination
Moret-Tatay, C., Baixauli-Fortea, I., & Grau-Sevilla, M. D. (2020). Profiles on the Orientation Discrimination Processing of Human Faces. International Journal of Environmental Research and Public Health, 17(16), 5772.
ii) and clinical ones
Storey, G. (2019). Deep human face analysis and modelling (Doctoral dissertation, Northumbria University, Newcastle upon Tyne, UK).

Response:
The authors agree with reviewer's opinion that the above-mentioned papers will broaden the scope and influence of FER research. Thus, Line 36 of the revised manuscript was modified as follows.
Old:
“FER has various applications, such as in-vehicle driver monitoring, human-robot interaction, psychology, and digital entertainment [5,6].”
New:
“FER has various applications, such as in-vehicle driver monitoring, facial modeling [3,4], psychology, and digital entertainment [5,6].”
References
[1] Liu, S.; Johns, E.; Davison, A. J. End-to-end multi-task learning with attention. In Proceedings of the IEEE Conference on Computer Vision and Pattern Recognition, Long Beach, CA, 16-20 June 2019, pp. 1871-1880.
[2] Yang, H.; Ciftci, U.; Yin, L. Facial expression recognition by de-expression residue learning. In Proceedings of the IEEE Conference on Computer Vision and Pattern Recognition, Salt lake City, Utah, 19-21 June, pp. 2168-2177.
[3] Thai Ly, S.; Do, N. T.; Lee, G. S.; Kim, S. H.; Yang, H. J. Multimodal 2D and 3D for In-the-wild Facial Expression Recognition. In Proceedings of the IEEE Conference on Computer Vision and Pattern Recognition Workshops, Long Beach, CA, 16-20 June 2019.
[4] Chen, S.; Wang, J.; Chen, Y.; Shi, Z.; Geng, X.; Rui, Y. Label Distribution Learning on Auxiliary Label Space Graphs for Facial Expression Recognition. In Proceedings of the IEEE Conference on Computer Vision and Pattern Recognition, Virtual, 14-19 June, pp. 13984-13993.

Round 2

Reviewer 1 Report

My questions in the first round review have been answered very well. The revised manuscript has been significantly improved. So I think that it can be accepted by Sensors.